

# Effects of exercise training with blood flow restriction on vascular function in adults: a systematic review and meta-analysis

Elisio A. Pereira-Neto[1,2], Hayley Lewthwaite[1,2,3], Terry Boyle[2,4], Kylie Johnston[1,2], Hunter Bennett[2,5] and Marie T. Williams[1,2]

[1] Innovation, IMPlementation And Clinical Translation in Health (IIMPACT), University of South Australia, Adelaide, South Australia, Australia
[2] Allied Health and Human Performance, University of South Australia, Adelaide, South Australia, Australia
[3] Department of Kinesiology and Physical Education, McGill University, Montreal, Quebec, Canada
[4] Australian Centre for Precision Health, University of South Australia, Adelaide, South Australia, Australia
[5] Alliance for Research in Exercise, Nutrition, and Activity (ARENA), University of South Australia, Adelaide, South Australia, Australia

Corresponding author
Elisio A. Pereira-Neto,
elisio.pereira_neto@mymail.unisa.edu.au

## ABSTRACT

**Background:** Blood flow restricted exercise (BFRE) improves physical fitness, with theorized positive effects on vascular function. This systematic review and meta-analysis aimed to report (1) the effects of BFRE on vascular function in adults with or without chronic health conditions, and (2) adverse events and adherence reported for BFRE.

**Methodology:** Five electronic databases were searched by two researchers independently to identify studies reporting vascular outcomes following BFRE in adults with and without chronic conditions. When sufficient data were provided, meta-analysis and exploratory meta-regression were performed.

**Results:** Twenty-six studies were included in the review (total participants $n = 472$; $n = 41$ older adults with chronic conditions). Meta-analysis ($k = 9$ studies) indicated that compared to exercise without blood flow restriction, resistance training with blood flow restriction resulted in significantly greater effects on endothelial function (SMD 0.76; 95% CI [0.36–1.14]). No significant differences were estimated for changes in vascular structure (SMD −0.24; 95% CI [−1.08 to 0.59]). In exploratory meta-regression analyses, several experimental protocol factors (design, exercise modality, exercised limbs, intervention length and number of sets per exercise) were significantly associated with the effect size for endothelial function outcomes. Adverse events in BFRE studies were rarely reported.

**Conclusion:** There is limited evidence, predominantly available in healthy young adults, on the effect of BFRE on vascular function. Signals pointing to effect of specific dynamic resistance exercise protocols with blood flow restriction (≥4 weeks with exercises for the upper and lower limbs) on endothelial function warrant further investigation.

## INTRODUCTION

Blood flow restricted exercise (BFRE) is low-intensity exercise performed with restriction of blood flow to the active muscle group (*Patterson et al., 2019*). Improvements in muscle mass and strength, sports performance and aerobic and functional capacity following BFRE have been reported to be similar to higher intensity traditional exercise (*Bennett & Slattery, 2018*; *Hughes et al., 2017*; *Pereira Neto et al., 2018*). It has been suggested that BFRE may, therefore, benefit populations that cannot perform traditional exercise at an intensity high enough for maximal physiological adaptations (for example, the elderly or people with chronic obstructive pulmonary disease (COPD) or osteoporosis) (*Hughes et al., 2017*; *Pereira Neto et al., 2018*; *Domingos & Polito, 2018*; *Patterson et al., 2017*). However, these populations often have comorbidities related to impaired vascular function (VF) (*Yoo et al., 2018*; *Pedralli et al., 2018*), which increase the risk of cardiovascular events and mortality (*Ambrosino et al., 2017*; *Chen et al., 2015*). From this perspective, it is important to understand the potential effects, positive or negative, of BFRE on VF.

Improvements in VF following participation in traditional exercise have been reported in systematic reviews (*Pedralli et al., 2018*; *Qiu et al., 2018*), with an increase in nitric oxide availability and reduction of oxidant enzymes proposed as underlying mechanisms (*Ashor et al., 2015*). The hypoxic conditions created during prolonged exercise with blood flow restriction, and the increased shear stress on the artery walls following release of the cuff, may increase the availability of nitric oxide and vascular endothelial growth factor, stimulating angiogenesis and consequently improving VF (*Horiuchi & Okita, 2012*). Conversely, it is possible that the accumulation of metabolites under anaerobic conditions during BFRE may trigger the metaboreflex, leading to increases in sympathetic activity and vasoconstriction, and greater total peripheral resistance (*Da Cunha Nascimento, Schoenfeld & Prestes, 2020*; *Cristina-Oliveira et al., 2020*). Studies that have investigated the effects of BFRE on VF have reported heterogenous results, with both beneficial and deleterious (*Karabulut, Karabulut & James, 2018*; *Paiva et al., 2016*; *Millis et al., 2017*) effects reported for VF.

Given this conflicting evidence, this systematic review aimed to explore the effects of BFRE on vascular function in adults with or without chronic health conditions. The specific question posed for this review was "In adults, compared to exercise training without blood flow restriction (non-BFRE), what effect does blood flow restricted exercise training (BFRE) have on vascular function?" Secondarily, we aimed to identify BFRE associated adverse events, adherence and attrition.

## SURVEY METHODOLOGY

### Search strategy

The review protocol was registered with the International Prospective Register of Systematic Reviews (PROSPERO, CRD42019147408) and reported according to the Preferred Reporting Items for Systematic reviews and Meta-Analyses (PRISMA) (*Moher, 2015*). Five electronic databases (PubMed, Web of Science, Scopus, Medline and EmBase) were searched for studies published from inception to 31$^{st}$ August 2019 and updated

on 19<sup>th</sup> August 2020, with no language restriction imposed. The following combination of terms were used within the search strategy: ("blood flow restriction" OR "blood flow occlusion" OR "vascular occlusion" OR "kaatsu training") AND ("vascular function" OR "endothelial function" OR "arterial stiffness" OR "pulse wave velocity" OR "flow-mediated dilatation" OR "VEGF1" OR "nitric oxide") (See Table S1 for example search strategy). The reference lists of relevant studies identified in the screening process were screened for any additional relevant studies.

## Eligibility criteria

Studies were eligible for inclusion in this review if they: (a) were a primary study published as full text in a peer-reviewed journal; (b) were a controlled trial with or without randomization (parallel groups, cross-over included), with at least one group including BFRE and, at least one group including non-BFRE (exercise of any form); (c) were conducted in adults (18 years and over with or without chronic health conditions); (d) applied blood flow restriction (BFR) using an external device to promote external pressure in the proximal part of a limb (e.g., cuff, elastic band); and (e) reported at least one outcome related to vascular function (e.g., flow-mediated dilatation (FMD), pulse-wave velocity (PWV), nitric oxide). Studies were excluded if they: (a) were single group pre-post studies comparing baseline outcomes with outcomes measured during or post-BFRE without a non-BFRE group comparator; or (b) used other forms of hypoxic training such as hyperbaric chamber or high-altitude training as the intervention group.

## Selection process

References identified from database searches were imported to systematic review software (Covidence systematic review software, Veritas Health Innovation, Melbourne, Australia) and exact duplicates removed. Title and abstract of references were initially screened for eligibility by two reviewers independently (EPN and MW). Where studies met eligibility or could not confidently be excluded, full text versions were accessed and screened by two reviewers independently (EPN and HB). Differences between reviewers were resolved by discussion and when consensus could not be reached, a third reviewer (MTW) was consulted. Hand searching of reference lists of included papers was undertaken to identify additional eligible studies.

## Data extraction

A data extraction template was developed and piloted a priori. Data extraction was performed by two reviewers independently (EPN and HB). Data extracted from studies included the following domains:

(a) Publication demographics (publication year, authors, country of data collection);

(b) Study details (design, sample size, participant age (mean, SD) and gender, health status (healthy or clinical condition));

(c) Exercise training regimen for intervention and control groups (protocol duration, exercise intensity, frequency and type, blood flow restriction protocol (intervention group), blood flow restriction pressure (intervention group));

(d) Primary outcomes: mean and standard deviation (SD) for vascular function outcomes pre- and post-intervention for the BFRE and non-BFRE groups, which could include but were not limited to: vascular structure (e.g., ankle-brachial index, arterial diameter, pulse-wave velocity), endothelial function (e.g., flow-mediated dilatation, resting blood flow, reactive hyperemia blood flow) and biomarkers (e.g., catalase, vascular endothelial growth factor, nitric oxide). For studies where the results for the primary outcomes were presented in figures, WebPlotDigitizer was used to extract data (*Rohatgi, 2019*);

(e) Secondary outcomes: adverse events (any unpleasant or unintended event, sign, symptom reported by authors as related to the intervention (*Australian National Health and Medical Research C, 2018*)), adherence (percentage of planned sessions attended (*Sohanpal et al., 2012*)) and attrition (proportion of randomized participants analyzed (*Sohanpal et al., 2012*)). Corresponding authors were contacted to seek for any missing primary or secondary outcomes.

## Study quality—appraisal of methodological bias

The methodological quality of the studies was assessed using the Tool for the Assessment of Study Quality and Reporting in Exercise (TESTEX) scale (validity and reliability intra-class correlation coefficient >0.91 (*Smart et al., 2015*)). This instrument consists of two sections (study quality and reporting). Standardized instructions guided scoring (minimum of 0 and maximum of 15 points), where higher scores indicated better study quality (0–5 points) and reporting (0–10 points) (*Sohanpal et al., 2012*). Assessment was performed by two reviewers independently (EPN and JP) for all eligible studies. Studies were not excluded based on score for methodological quality as this has been shown to reduce precision and increase bias on meta-analysis results (*Stone et al., 2019*; *Higgins et al., 2011*).

## Data management and analysis

### Descriptive analysis

Details on study publication and protocol characteristics were tabulated and a narrative synthesis was used to summarize study demographics (author, publication year), design, sample characteristics, exercise training regimens and primary and secondary outcomes. Outcomes for vascular function were grouped under three broad constructs: 'Vascular structure', 'Endothelial function' and 'Biomarkers'. Frequency and percentage were used to present data regarding adverse events, adherence and attrition across studies. When adverse events were reported, the number of participants affected, and verbatim description of adverse events were collated.

### Meta-analysis

A meta-analysis was performed where studies were comparable in design, intervention duration and outcomes for Vascular structure and Endothelial function. Meta-analysis was not performed for the Biomarker construct as measures within this construct represent a diverse range of aspects related to vascular function and it was deemed not

appropriate to combine them for meta-analysis. For studies that reported more than one outcome within the Vascular structure and Endothelial function constructs, PWV (vascular structure) and FMD (endothelial function) were chosen as the outcome of interest as these are considered the gold standard (*Tousoulis, Antoniades & Stefanadis, 2005*). In the absence of studies reporting PWV or FMD, the outcome with the most rigorous validation according to available best evidence, or the most commonly reported outcome across studies, was chosen. As the Ankle-Brachial index (ABI) has a preferred range of normality rather than a direction for improvement (*Resnick et al., 2004*), studies reporting ABI as a measure of vascular structure were not included in the meta-analysis.

The standardized mean difference (SMD) was calculated using the post-intervention outcome mean, standard deviation (SD) and sample size for the BFRE (experimental) and non-BFRE (control) groups. When these metrics were not reported, authors were contacted, or SD was calculated from the standard error reported (*Higgins et al., 2019*). If a study had more than one control group (e.g., high- and low-intensity non-BFRE), separate meta-analyses were undertaken (e.g., BFRE *versus* high-intensity non-BFRE and BFRE *versus* low-intensity non-BFRE). Heterogeneity between studies was analyzed with $I^2$ (with values stablished as $I^2 = 25\%$ (low), $I^2 = 50\%$ (moderate) and $I^2 = 75\%$ (high)) (*Grant & Hunter, 2006*) and Cochran's Q test (significance $p < 0.05$). Random-effects meta-analyses were used to compare BFRE and non-BFRE groups due to the different characteristics of the interventions used (*Rodrigues & Ziegelmann, 2010*; *Borenstein et al., 2010*). Meta-analyses were conducted using RevMan (Review Manager Version 5.3, The Cochrane Collaboration, 2014).

### Meta-regression

Meta-regression was used to explore associations between BFRE protocol factors and effect sizes of vascular structure and endothelial function outcomes in the BFRE group of included studies. For studies that reported more than one outcome per construct, the outcome of interest was chosen using the same rationale described above. Predictor variables explored were: (a) study design (randomized controlled trial or randomized cross-over trial with controlled trial as reference); (b) population group (healthy older adults or clinical population with healthy adults as reference); (c) exercise modality (treadmill walking, dynamic resistance training or others with handgrip as reference); (d) exercised limbs (upper or lower limbs with both limbs as reference); (e) length of intervention (≥4 weeks with single session as reference); (f) number of sets (3, 4 or ≥5 with ≤2 as reference); (g) cuff width (continuous variable); (h) approach to determine restriction pressure (arterial occlusion pressure, systolic blood pressure or limb circumference, with pre-determined as reference); (i) mean restriction pressure (continuous variable); and methodological quality (TESTEX score, continuous variable). Associations between predictor variables and outcome effect sizes were explored when there were ≥10 studies that reported on an outcome within the construct (vascular structure or endothelial function) (*Higgins et al., 2019*). Effect sizes and variance were calculated using pre- and post-intervention mean pre-intervention SD, and pre-post

correlation coefficient ($r$). When these metrics were not reported, we obtained this information by (in order or preference): contacting authors; calculating SD standard error or confidence interval and/or calculating $r$ using the mean and SD of change from pre to post (*Higgins et al., 2019*); or using the mean of $r$ from all other studies (*Higgins et al., 2019*). Random-effects meta-regression models were conducted separately for each outcome construct. Meta-regression was performed in R Studio 1.2.5 (RStudio Inc., Boston, MA, USA) using the package 'metafor'. A two-sided significance level of $p < 0.05$ was used for all analyses.

## RESULTS

The search identified 1,047 references, of which 26 studies were eligible for inclusion in this review (See Fig. S1).

### Study design and population

The majority of included studies were controlled trials (13 randomized controlled trials (RCT), 10 randomized cross-over trials, two non-randomized controlled trials and a case report; see Table S2). There were 15 (58%) studies that evaluated BFRE *versus* one other group, six studies had two comparator groups and five studies had three comparator groups (See Table S2). Across the 26 studies, 472 individuals (303 males, 169 females, aged 18 to 91 years) were included. The majority of studies ($n = 23$, 88%) were in non-clinical populations (total $n = 431$, 284 males, 147 females, aged 18 to 81 years). Three studies recruited participants with chronic conditions (total $n = 41$; 19 males, 22 females; aged 58 to 91 years), with each study representing a different clinical condition (coronary artery disease, hypertension, sarcopenia). Overall, methodological quality was moderate (median 9.5) with scores ranging between 6–13 (Details presented in Table S3).

### Comparators

The most frequent comparators to the active intervention (low-intensity exercise with BFR) were exercise training without BFR at low ($n = 17$, 65%) or higher ($n = 8$, 31%) exercise intensities. Two studies included four groups (with and without BFR at low- and moderate-intensity exercise). Single studies compared different (i) restriction pressures (40% and 80% of maximum occlusion pressure) or (ii) BFR position (upper and lower limbs) with low and high intensity exercise to low and high intensity exercise without BFR. The study that used neuromuscular electrical stimulation (NMES) compared the same intensity of maximum voluntary contraction (MVC) with and without BFR.

### Blood flow restricted exercise protocols

#### Exercise mode

In the majority of studies, BFR was combined with low-intensity dynamic resistance exercise ($n = 18$, 69%). Exercise of the lower limbs was performed in 12 (46%) studies, upper limbs in seven studies (27%) and both upper and lower limbs in seven studies (27%). Intervention duration varied across studies from a single exercise session ($n = 10$, 39%) to training durations of 4 and 16 weeks with two to three exercise sessions per week ($n = 14$,

54%). Two studies included assessment of a single exercise session preceding an intervention of 4 and 6 weeks (See Table S4).

### Exercise intensity

The approach to prescribe exercise intensity differed according to the type of exercise training (See Table S4). For dynamic resistance exercise, the most common method was the one-repetition maximum strength test (1RM) ($n = 15$; 58%, exercise intensity range 20% to 40% of 1RM). For isometric resistance exercise ($n = 4$, 15%) and NMES ($n = 1$, 4%), training intensity was based on a percentage of maximum voluntary contraction (MVC; intensity range between 10% to 60% MVC).

### Blood flow restriction (BFR)

All studies used cuffs to achieve BFR (pneumatic or hand-pumped) and, where reported ($n = 18$, 69%), cuff width ranged between 3 to 23 cm. The selection of restriction pressure was based on a variety of criteria: predetermined non-individualized pressure ($n = 14$, 54%), percentage of systolic blood pressure (SBP) ($n = 8$, 31%), percentage of limb occlusion pressure (LOP) ($n = 2$, 8%), limb circumference ($n = 1$, 4%) and one study used both SBP and LOP criteria depending on whichever was higher, with values ranging from 80 to 220 mmHg.

## Primary outcomes

Table 1 presents a summary of outcomes for Vascular structure, Endothelial function and Biomarkers.

### Vascular structure

Across 15 studies, six different outcomes were reported for vascular structure, with vascular compliance being the most frequently reported ($n = 6$, 23%). Of the 15 studies reporting vascular structure outcomes, 13 (87%) reported no differences between exercise with and without BFR (Table 1). Significant between group differences in vascular structure were reported in two studies following an exercise intervention of 16 weeks of BFR with resistance training (compared to high-intensity non-BFRE) (*Ozaki et al., 2013*) or a single session of whole-body vibration training with BFR (compared to whole-body vibration without BFR) (*Karabulut, Karabulut & James, 2018*).

### Endothelial function

Fourteen studies reported six different outcomes for endothelial function (FMD most frequently reported, $n = 8$, 31%). Of these 14 studies, eight (57%) reported no significant difference in endothelial function outcomes between exercise with and without BFR. For FMD, compared to non-BFRE, improvements were reported following four weeks of BFRE (*Renzi, Tanaka & Sugawara, 2010*), a single session of BFRE (walking) (*Credeur, Hollis & Welsch, 2010*) and 2 weeks of BFRE using handgrip training (*Paiva et al., 2016*). Blood flow (resting or reactive hyperemia) was reported to be improved after BFRE, compared to low-intensity non-BFRE, in healthy younger (*Patterson & Ferguson, 2010*) and older adults (*Patterson & Ferguson, 2011*; *Shimizu et al., 2016*) and a single subject with sarcopenia (*Lopes et al., 2019*). In studies reporting tissue oxygenation (*Shimizu et al.,*
**Table 1 Summary of vascular function outcomes assessed in the studies and differences between BFR and non-BFR conditions.**

| First author year | Vascular structure | | | | | | Endothelial function | | | | | | Biomarkers | | | | | | |
|---|---|---|---|---|---|---|---|---|---|---|---|---|---|---|---|---|---|---|---|
| | ABI | Arterial diameter | PWV | CAVI | Vascular compliance | Vascular conductance | FMD | RBF | RHBF | Dilatory capacity | RHI | TpO$_2$ | vWF | VEGF$_1$ | NO | GH | FDP | SOD | CAT |
| Ramis 2020 | | | | | | | = | | | | | | | | = | | | | |
| Credeur 2019 | | | = | | | | = | | = | | | | | | | | | | |
| Kambic 2019 | | | | | | | = | | | | | | | | | | | | |
| Lopes 2019 | | | | | | | | | (+) | | | | | | | | | | |
| Mouser 2019 | | | | = | | | | | | | | | | | | | | | |
| Barili 2018 | | | | | | | | | | | | | | | = | | | (+) LI/ = HI | |
| Boeno 2018 | | | | | | | | | | | | | | | = | | | (+) LI/= HI | (+) LI/= HI |
| Karabulut 2018 | | | | | (−) | | | | | | | | | | | | | | |
| Natsume 2018 | | | | | | | | | | | | | | | | (+) | | | |
| Sardeli 2017 | | | | | = | | | | | | | | | | | | | | |
| Paiva 2016 | | = | | | | | (−) | | | | | | | | | | | | |
| Shimizu 2016 | | | | | | | | | = | | (+) | | (+) | = | | (+) | | | |
| Yasuda 2016 | | | | = | | | | | | | | | | | | | = | | |
| Yasuda 2015a | | | = | = | | | = | | | | | | | | | | = | | |
| Yasuda 2015b | | | = | = | | | = | | | | | | | | | | = | | |
| Fahs 2014 | | | = | | = | = | | | | | | | | | | | | | |
| Ozaki 2013 | | | | | (+) | | | | | | | | | | | | | | |
| Fahs 2012 | | | | | = | = | | | = | | | | | | = | | | | |
| Hunt 2012 | | | | | | | = | | = | = | | | | | | | | | |
| Larkin 2012 | | | | | | | | | | | | = | | = | | | | | |
| Clark 2011 | | | = | | | | = | | | | | | | | | | | | |
| Figueroa 2011 | | | = | | | | | (+) | | | | | | | | | | | |
| Patterson 2011 | | | | | | | (−) | (+) | (+) | | | | | | | | | | |
| Credeur 2010 | | = | = | | | | | | | | | | | | | | | | |
| Patterson 2010 | | | | | | | | (+) LI = MI | (+) LI = MI | | | | | | | | | | |
| Renzi 2010 | = | = | | | | | (−) | = | | | | | | | | | | | |

**Note:**

BFR, Blood flow restriction; HI, High intensity; MI, Moderate intensity; LI, Low-intensity; ABI, Ankle-brachial index; PWV, Pulse wave velocity; CAVI, Cardio-ankle vascular index; FMD, Flow-mediated dilatation; RBF, Resting blood flow; RHBF, Reactive hyperemia blood flow; RHI, Reactive hyperemia index; TpO$_2$ Tissue oxygenation; vWF, von Willebrand factor; VEGF$_1$, Vascular endothelial growth factor; NO, Nitric oxide; GH, Growth hormone, FDP, Fibrin degradation products; SOD, Superoxide dismutase; CAT, Catalase; =, No statistically significant difference between conditions with and without BFR; (+), BFR condition promoted significant positive changes when compared to non-BFR; (−), BFR condition promoted significant negative changes when compared to non-BFR.

**Figure 1 Forest plot presenting comparison of vascular structure between high intensity non-BFR training and low intensity BFRE.** Values for PWV were reversed to meet direction of vascular compliance where increase means improvement; Squares represent individual study Hedges' g with size corresponding to weight of study and lines are 95% confidence intervals (CI); diamond indicates overall Hedges' g with its width corresponding to 95% CI.

*2016*; *Larkin et al., 2012*) or dilatory capacity (*Hunt, Walton & Ferguson, 2012*), no significant differences were reported between exercise with and without BFR.

### Biomarkers

Ten studies reported seven different biomarker outcomes, with nitric oxide (NO) being the most frequently reported ($n = 4$, 15%). Of these 10 studies, six (60%) studies reported no difference in biomarker outcomes between exercise with and without BFR. Growth hormone ($n = 2$) was increased in the BFRE condition compared to low intensity non-BFRE in healthy adults following a single session of handgrip training (*Natsume et al., 2018*) and in older individuals after four weeks of dynamic resistance exercise (*Shimizu et al., 2016*). Superoxide dismutase (*Barili et al., 2018*; *Boeno et al., 2018*) ($n = 2$, 8%) and catalase (*Boeno et al., 2018*) ($n = 1$, 4%) showed higher concentrations in the BFRE condition following a single session of treadmill walking in hypertensive older women (*Barili et al., 2018*) and in healthy adults after dynamic resistance exercise (*Boeno et al., 2018*).

## Meta-analysis

Nine studies were sufficiently homogenous (design, training protocol and population) for inclusion in the meta-analysis (*Ozaki et al., 2013*; *Patterson & Ferguson, 2010*, *2011*; *Shimizu et al., 2016*; *Clark et al., 2011*; *Fahs et al., 2012*; *Mouser et al., 2019*; *Yasuda et al., 2015a*; *Yasuda et al., 2015b*). All studies included in the meta-analysis were in non-clinical populations and explored the effects of BFR with isotonic dynamic resistance training for four weeks or more. Separate models were run comparing the effects of low intensity BFRE to high intensity non-BFRE on vascular structure, and low intensity BFRE to low intensity non-BFRE on endothelial function.

### Low intensity BFRE versus high intensity non-BFRE—Vascular structure

Four studies reported outcomes for vascular structure (vascular compliance ($n = 3$), (*Ozaki et al., 2013*; *Fahs et al., 2012*; *Mouser et al., 2019*); and PWV ($n = 1$) (*Clark et al., 2011*)) after four or more weeks of dynamic resistance training (low-intensity BFRE *versus* high-intensity non-BFRE) (Fig. 1). No significant difference was observed between low intensity BFRE and high intensity non-BFRE (SMD -0.24, 95% CI [−1.08 to 0.59], $p = 0.57$) (Fig. 1). Moderate and significant heterogeneity was observed ($I^2 = 73\%$, $p = 0.01$).

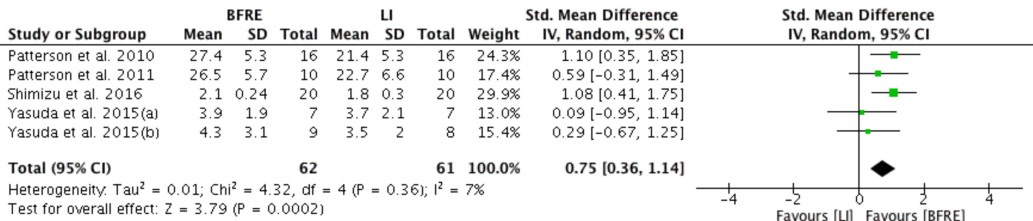

**Figure 2 Forest plot presenting comparison of endothelial function between low intensity non-BFR training and low intensity BFRE.** Increase means improvement for all outcomes; Squares represent individual study Hedges' g with size corresponding to weight of study and lines are 95% confidence intervals (CI); diamond indicates overall Hedges' g with its width corresponding to 95% CI.

### Low intensity BFRE versus low intensity non-BFRE—Endothelial function

Five studies provided sufficient data for meta-analysis of the effect of four or more weeks of low intensity dynamic resistance exercise training with and without BFR on endothelial function (Fig. 2). Specific endothelial outcomes were FMD ($n = 2$) (*Yasuda et al., 2015a, 2015b*), reactive hyperemia blood flow ($n = 2$) (*Patterson & Ferguson, 2010, 2011*) and reactive hyperemia index ($n = 1$) (*Shimizu et al., 2016*). Meta-analysis estimated low-intensity dynamic resistance training with BFR had a significant positive difference compared to low intensity exercise without BFR (SMD 0.76, 95% CI [0.36 to 1.14], $p < 0.001$) for endothelial function (Fig. 2). Heterogeneity was considered low ($I^2 = 7\%$) and not significant ($p = 0.36$).

### Meta-regression

Findings from meta-regression analyses are summarized in Table S5 (*Lopes et al. (2019)* excluded as case study). For vascular structure, 15 studies were included within meta-regression models reflecting four vascular structure outcomes: pulse-wave velocity (PWV) ($n = 5$) (*Credeur, Hollis & Welsch, 2010*; *Clark et al., 2011*; *Credeur et al., 2019*; *Fahs et al., 2014*; *Figueroa & Vicil, 2011*), vascular compliance ($n = 5$) (*Karabulut, Karabulut & James, 2018*; *Ozaki et al., 2013*; *Fahs et al., 2012*; *Mouser et al., 2019*; *Sardeli et al., 2017*), cardio-ankle vascular index ($n = 3$) (*Yasuda et al., 2015a*; *Yasuda et al., 2015b*; *Yasuda et al., 2016*) and arterial diameter ($n = 2$) (*Paiva et al., 2016*; *Renzi, Tanaka & Sugawara, 2010*). No significant associations were identified between BFRE protocol factors and effect sizes (all $p$-values >0.05).

For endothelial function, 13 studies were included in the meta-regression reflecting three outcomes: FMD ($n = 8$) (*Paiva et al., 2016*; *Renzi, Tanaka & Sugawara, 2010*; *Hunt, Walton & Ferguson, 2012*; *Yasuda et al., 2015a*; *Yasuda et al., 2015b*; *Credeur et al., 2019*; *Ramis et al., 2020*; *Kambic et al., 2019*), reactive hyperemia blood flow ($n = 4$) (*Patterson & Ferguson, 2010, 2011*; *Credeur et al., 2019*; *Fahs et al., 2014*) and reactive hyperemia index ($n = 1$) (*Shimizu et al., 2016*). Study design was significantly associated with effect size, with randomized cross-over trials associated with smaller effect sizes compared to control trials (regression estimate (ß) of −2.91; 95% CI [−5.40 to −0.42]; $p = 0.02$). For exercise modality, compared to handgrip training, treadmill walking (ß = −2.62; 95% CI [−4.18 to −1.06]; $p = 0.001$) and dynamic resistance training

(ß = 1.64; 95% CI [0.83–2.45]; $p < 0.001$) were associated with significantly smaller and larger effects on endothelial function, respectively. Upper limb only exercise was associated with smaller effect sizes (ß = −2.22; 95% CI [−4.23 to −0.19]; $p = 0.03$) compared to exercise of the lower and upper limbs combined. For the number of exercise sets, compared to studies of ≤2 sets, exercise protocols with three (ß = 1.66; 95% CI [0.21–3.11]; $p = 0.02$) and four (ß = 1.54; 95% CI [0.03–3.05]; $p = 0.05$) sets were associated with larger effect sizes, while protocols with five or more sets (ß = 2.5; 95% CI [−4.69 to −0.30]; $p = 0.03$) were associated with smaller effect sizes. Meta-regression analysis for length of intervention was performed using 'single session' studies as the reference for duration, with durations explored as a continuous variable (number of weeks, $p > 0.05$) or categorical variable of ≥4 weeks (e.g., all studies with 4 weeks or more grouped in one category). Compared to a single session, interventions of ≥4 weeks were associated with larger effect sizes (ß = 1.92; 95% CI [0.39–3.46]; $p = 0.01$).

### Secondary outcomes

Table S6 presents a summary of adverse events, adherence, and attrition rates. The majority of studies ($n = 18$, 69 %) did not report the presence or absence of adverse events. Seven studies explicitly reported an absence of adverse events ($n = 7$, 27%). A single study reported an initial sensation of leg numbness in two participants once they commenced BFRE, which did not persist beyond the first minute of exercise (*Barili et al., 2018*) (2 out of 16 hypertensive older women). Adherence and attrition were reported in 11 (42%) and six (23%) studies, respectively. Adherence was reported or calculated as 100% for 11 studies and dropout rates ranged from 5% to 40%.

## DISCUSSION

This systematic review aimed to report BFRE effects on vascular function in adults with or without chronic health conditions and, secondarily, to identify BFRE associated adverse events, adherence and attrition. To the best of our knowledge, this systematic review presents the first quantitative analyses exploring differences between BFRE and non-BFRE on vascular function, and the first exploratory meta-regression model to explore the influence of BFRE protocol components on vascular function. We found that the body of evidence is modest, heterogenous and predominantly included apparently healthy participants. With these caveats in mind, the main findings of this review were: (a) a variety of outcomes were used to assess vascular function in the domains of vascular structure, endothelial function and biomarkers; (b) compared to non-BFRE, meta-analysis favored dynamic resistance exercise with BFR for endothelial function outcomes (no difference between interventions for vascular structure outcomes); (c) study design, mode of exercise, training volume and length of exercise intervention were predictors of endothelial function effect size with BFRE training; (d) adverse events in BFRE are potentially underreported.

Previously, BFRE research has focused on muscle strength and size (*Lixandrão et al., 2018*) and aerobic capacity (*Silva et al., 2019*), with rising interest in other aspects such as blood pressure (*Domingos & Polito, 2018*) and perceptual measures (*Spitz et al., 2020*).

This review suggests that the body of evidence for BFRE and VF is small, with heterogeneous exercise protocols and predominantly in healthy individuals. Interest is increasing concerning potential use of BFRE within older clinical populations (e.g., cardiovascular disease and chronic lung conditions) (*Hughes et al., 2017*). There appears to be little information available concerning efficacy and safety of BFRE and impact upon VF specific to people living with chronic conditions—a clinical population where VF is likely impaired. We considered VF as a number of aspects related to vessel structure, functioning of the endothelium and substances which regulate both structure and functioning. This approach was required as few studies reported use of the same outcome measure (e.g., FMD was the most common outcome but reported by approximately one third of studies), which limited our ability to perform more robust analysis (*Higgins et al., 2019*).

## BFRE and vascular structure

Our meta-analysis did not identify significant differences between BFRE and non-BFRE for arterial stiffness outcomes. Increased sympathetic nervous system activity is a common concern with BFRE (*Da Cunha Nascimento, Schoenfeld & Prestes, 2020*). The reduction of blood flow to the muscle increases vascular peripheral resistance and metabolite concentration, which may trigger baro- and chemoreceptors that stimulate sympathetic outflow (*Renzi, Tanaka & Sugawara, 2010*). Domingos and Polito (2018)'s meta-analysis reported BFRE to cause greater increases in blood pressure than low intensity traditional resistance training (*Domingos & Polito, 2018*), indicating exercise-induced increases in sympathetic activity, which might be expected to increase arterial stiffness and negatively affect vascular structure. The same meta-analysis (*Domingos & Polito, 2018*) and previous randomized trials (*Cezar et al., 2016*; *Araújo et al., 2014*) have reported a more pronounced post-exercise hypotensive effect for BFRE compared to high- and low intensity traditional resistance training, which is indicative of reduced sympathetic activity (*Cristina-Oliveira et al., 2020*). Despite greater sympathetic activity with BFRE compared to low intensity exercise without BFR, the post-exercise reduction of sympathetic outflow associated with BFRE could potentially act as a protector of the vascular system.

## BFRE and endothelial function

In healthy individuals, our meta-analysis indicated that four or more weeks of low intensity dynamic resistance exercise with BFR was associated with greater improvements in endothelial function compared to low intensity non-BFRE. Increased shear stress is hypothesized to be a modulator of chronic endothelial function changes following exercise (*Pedralli et al., 2018*; *Paiva et al., 2016*). Improvements in FMD following traditional moderate-to-high dynamic resistance exercise have been proposed to be associated with the process of intermittent blood flow in the small muscular arteries resulting from muscle contraction, increasing shear stress in the artery walls and promoting release of vasodilator factors (i.e., NO, VEGF, GH) (*Paiva et al., 2016*). It is possible that hypoxic conditions created during prolonged exercise with blood flow occlusion and/or the

increased shear stress following release of the cuff, increase the availability of vasodilator factors consequently improving endothelial function.

## BFRE and vascular function biomarkers

Few studies included in this review ($n = 10$; 35%) reported VF biomarker outcomes (*Ozaki et al., 2013*; *Shimizu et al., 2016*; *Larkin et al., 2012*; *Natsume et al., 2018*; *Barili et al., 2018*; *Boeno et al., 2018*; *Yasuda et al., 2015a*, *2015b*; *Yasuda et al., 2016*; *Ramis et al., 2020*). Within these studies, seven different outcomes were reported, which in included outcomes not unique to vascular function (i.e., growth hormone (*Natsume et al., 2018*)). Only four of these studies reported between group differences for VF biomarkers (*Shimizu et al., 2016*; *Natsume et al., 2018*; *Barili et al., 2018*; *Boeno et al., 2018*) and, while all four favored BFRE, these studies differed in design, restriction pressure, exercise regimen, exercise intensity and assessed different outcomes (growth hormone, von Willebrand factor, catalase and superoxide dismutase) making it difficult to draw conclusions on the overall effects of BFRE on vascular function biomarkers. Given the variety of outcomes observed in this review, and the heterogeneity of study designs/protocols, it is not clear which biomarkers should be considered as priorities for studies investigating BFRE and vascular function.

## Which characteristics of BFRE protocols were associated with vascular function or structure effects sizes? Findings of the meta-regression

Our meta-regression results should be interpreted with caution due to the small number of studies (total k = 15 for vascular structure and k = 13 for endothelial function, with fewer common studies for each specific outcome within the broader domains), high heterogeneity between study protocols, and recognition that within each bivariate model, effect sizes for each variable and differences between strength of association are compared with a reference variable, negating identification of the causal influence of different predictors (*Higgins et al., 2019*). In addition, only four (16%) studies reported a sample size estimate, making it difficult to determine whether the remaining studies ($n = 22$; 84%) were adequately powered. Underpowered studies may provide inflated effect sizes and reduce the chance of detecting a true difference (*Button et al., 2013*; *Turner, Bird & Higgins, 2013*). With these forewarnings in mind, a number of protocol 'signals' were evident for endothelial function effects (but not vascular structure).

Intervention duration as a categorical, but not a continuous variable, was a significant predictor of effect size. Endothelial function measures are prone to change after a single bout of exercise (*Ashor et al., 2015*), the degree of which appears to differ between exercise type (*Ashor et al., 2015*). As such, the time frame required for physiological changes to endothelial function in response to chronic exercise varies. It is likely that other BFRE protocol factors (e.g., approach to determine pressure, cuff width) influence the relationship between intervention duration and endothelial function, which could only be detected in a multivariate analysis.

Consistent with known exercise training principles (*Schoenfeld, 2013*), we found exercise volume and mode (engaging larger muscle groups) to be important for endothelial function outcomes. Previous meta-analysis on non-BFRE (i.e., traditional exercise training) identified higher exercise volumes, compared to lower exercise volumes, resulted in greater improvements in skeletal muscle hypertrophy (*Schoenfeld, Ogborn & Krieger, 2017*), potentially caused by an increased release of neurohumoral factors (i.e., GH, nitric oxide, vascular endothelial growth factor). These same neurohumoral factors are known to modulate endothelial function (*Schoenfeld, 2013*). It is, therefore, possible that higher training volumes will also impact vascular outcomes. Within our meta-regression, there were also variables signaling potential harm – albeit from a single study. The association between training at five or greater sets per exercise and smaller effect sizes for endothelial function may reflect the concept that elevated training volumes can produce higher inflammatory responses (*Calle & Fernandez, 2010*) and, consequently, decrease endothelial function. This is in agreement with a recent review of BFRE methodology, application and safety which recommended BFR training use dynamic resistance exercise of four sets per exercise (*Patterson et al., 2019*).

Within our review, reported cuff widths ranged between 3 to 23 cm ($n = 18$) and the most common approach reported for determining restriction pressure was use of a predetermined, non-individualized pressure ($n = 13$) with systolic blood pressure ($n = 10$) and arterial occlusion pressure ($n = 3$) accounting for the remaining approaches. The approach used to determine restriction pressure (e.g., predetermined non-individualized, systolic blood pressure or LOP) did not significantly predict of effect size for arterial structure or endothelial function. The review published by *Patterson et al. (2019)* recommended using a percentage of an individual's LOP. The recency of this recommendation (published May 2019) is likely to explain why the majority (92%) of studies did not use LOP to prescribe restriction pressure. Limitations of alternative approaches to determining BFRE restriction training pressures include inter limb/individual variation in limb circumferences necessitating different cuff pressures and inconsistent associations between blood pressure and the pressure required to achieve occlusion (*Loenneke et al., 2012*). Higher restriction pressures (i.e., non-individualized 220 mmHg, which is 100 mmHg higher than normal SBP) also appear to increase discomfort and cardiovascular responses (*Jessee et al., 2017*), which may increase the risk of adverse vascular events (*Loenneke et al., 2012*). Future studies in BFRE and vascular function are more likely to individually prescribe training pressure using a percentage of LOP.

Blood pressure responses following BFRE have been explored in prior meta-analysis (23 studies included in the analysis of normotensive and hypertensive individuals (*Domingos & Polito, 2018*)), with hypertensive participants showing higher blood pressure changes. There is very little data available on the effects of BFRE in the vascular function of older individuals with chronic conditions. Our review included 41 individuals (of the total $n = 472$) with chronic conditions, and the review by *Domingos & Polito (2018)* included 95 (of the total $n = 325$). As populations living with chronic conditions often

present with increased arterial stiffness and endothelial dysfunction (*Chen et al., 2015*), it is likely that they will have different vascular function responses to BFRE than healthy adults. Notably, the majority of studies included within this review excluded participants with comorbidities such as peripheral vascular disease (assessed by the ankle-brachial index) (*Loenneke et al., 2012*). The findings of this review concerning the effects of BFRE on vascular function should therefore not be generalized to older populations living with chronic cardiovascular comorbidities.

## Secondary outcomes

In a survey (*Patterson & Brandner, 2017*) of 115 practitioners who used BFRE, 39% ($n = 60$) reported at least one side effect observed in their practice. Whereas in *Hughes et al. (2017)*'s review of BFRE and our review, 50% and 69% of studies (respectively) made no mention of adverse events. In these studies, it is not clear if adverse event did not occur or if they were not assessed or reported. The potential underreporting of either presence or absence of adverse events makes it difficult to determine the safety of BFRE and transparent reporting should be considered a priority in future BFRE studies.

Strengths of this review include a comprehensive search of major databases related to the review topic, with no language or time restrictions, and a rigorous approach to conducting the review and reporting of results. Due to heterogeneity of exercise regimens and outcomes in the studies included, it was not possible to perform adjustment for potential confounding factors in the meta-regression. Publication bias was not assessed as the funnel plot tests to detect asymmetry are compromised in meta-analysis with small numbers (<10) of studies (*Higgins et al., 2019*). Translation of our results to clinical populations is limited as the majority of included studies were in healthy individuals.

## CONCLUSIONS

BFRE has been suggested as potentially useful in older clinical populations unable to train at the intensity and duration essential for physiological responses. However, such populations may also have altered vascular function. This review identified a small, heterogeneous evidence base for the effect of BFRE on vascular function/structure with studies prominently including younger healthy individuals. Meta-analysis findings suggested BFRE training to have a positive effect compared to non-BFRE on endothelial function, but not on vascular structure. There are indications that BFRE protocol factors, such as the use of dynamic resistance training, ≥4 weeks duration and exercise protocols which include training for both upper and lower limbs, are associated with larger effect sizes for vascular function outcomes.

For clinicians/researchers planning future studies of BFRE, consideration should be given to recruiting older people with and without chronic conditions, including outcomes for vascular function, adverse events and tolerability and adopting recommendations from the review by *Patterson et al. (2019)* for determining restriction pressure and prescribing exercise, as this will standardize protocols and facilitate comparison of findings.

**Key findings:**

- The evidence regarding the effects of blood flow restricted exercise (BFRE) training in vascular function is small and heterogeneous.
- Our results suggest that BFRE training may have a positive effect on endothelial function and that BFRE protocol factors (e.g., mode and duration of exercise) are associated with larger effect sizes for vascular function outcomes.
- It is necessary to report adverse events and include vascular function outcomes in future BFRE training studies to clarity its safety.

### Funding
Elisio A. Pereira-Neto was supported by a University of South Australia and Australian Technology Network PhD scholarship funding. The funders had no role in study design, data collection and analysis, decision to publish, or preparation of the manuscript.

### Grant Disclosures
The following grant information was disclosed by the authors:
Elisio A Pereira-Neto was supported by a University of South Australia and Australian Technology Network PhD scholarship funding. The funders had no role in study design, data collection and analysis, decision to publish, or preparation of the manuscript.

### Competing Interests
The authors declare that they have no competing interests.

### Author Contributions
- Elisio A. Pereira-Neto conceived and designed the experiments, performed the experiments, analyzed the data, prepared figures and/or tables, authored or reviewed drafts of the paper, and approved the final draft.
- Hayley Lewthwaite conceived and designed the experiments, prepared figures and/or tables, authored or reviewed drafts of the paper, and approved the final draft.
- Terry Boyle conceived and designed the experiments, analyzed the data, prepared figures and/or tables, authored or reviewed drafts of the paper, and approved the final draft.
- Kylie Johnston conceived and designed the experiments, prepared figures and/or tables, authored or reviewed drafts of the paper, and approved the final draft.
- Hunter Bennett conceived and designed the experiments, performed the experiments, analyzed the data, prepared figures and/or tables, authored or reviewed drafts of the paper, and approved the final draft.
- Marie T. Williams conceived and designed the experiments, prepared figures and/or tables, authored or reviewed drafts of the paper, and approved the final draft.

## Data Availability

The raw data used for statistical analysis (meta-analysis and meta-regression) are available as a Supplemental File.

## Supplemental Information

Supplemental information for this article can be found online at http://dx.doi.org/10.7717/peerj.11554#supplemental-information.

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
