# Peer review of "Effects of exercise training with blood flow restriction on vascular function in adults: a systematic review and meta-analysis"

_PeerJ, doi:10.7717/peerj.11554_

## Round 0.1 · original submission · Minor Revisions

· Academic Editor

Minor Revisions

Thank you for a well written manuscript. Please address the comments and suggestions in detail.

Reviewer 1 ·

Basic reporting

No comment

Experimental design

No comment

Validity of the findings

No comment

Additional comments

The authors set out to write a systematic review and meta-analysis over the effects of blood flow restricted exercise on vascular function in humans.

The authors have done very well in following all of the best-practice guidelines as they relate to conducting such reviews and analyses. I have read the manuscript several times over, and I cannot find anything material that I feel absolutely must be added to the manuscript.

I am in agreement with the articles that have been included in this review and analysis, and all included articles meet the authors' stated criteria for inclusion. The meta-analyses appear to be performed correctly. The writing is clear and concise.

I appreciate the authors' inclusion of careful wording with regard to the wide variety of BFR protocols, the limited number of studies in this particular research area, and that all results of these analyses should be viewed cautiously as a result. Ideally, this review will allow other laboratories to focus their expertise on some of the gaps in the literature that the authors noted.

·

Basic reporting

The authors of this manuscript did a very nice job on reporting the information retrieved from their hypotheses and presenting the information in numerous figures/tables that make it easy for the reader to understand the results. Their raw data was reported and appears to reflect the analyses performed in the MA.

Experimental design

The authors did a commendable job in providing context for the experimental question and appropriately narrowed their inclusion criteria to reflect the BFR literature. They identified current gaps in the literature and where this manuscript would seek to address these gaps. Methods are appropriately described and depicted in figure 1 to allow for reproduction.

Validity of the findings

The authors provided comprehensive depictions of the data retrieved in Tables 1-5. They provide appropriate conclusions as to what can be determined based on the current body of evidence and provide future investigators some context/guidance to promote future homogeneity in the newer research being conducted on BFRE and its effects on vascular structure/function. They also accurately describe the caution to the reader on applying their results outside of the major populations studied (ie limited generalizability to clinical populations but more relevant for healthy populations).

Additional comments

The authors of this manuscript performed a very comprehensive systematic review and meta-analysis on the effects of BFRE on vascular structure and function along with relevant secondary outcomes (adherence/negative side effects). The authors should be commended at the level of data synthesis performed in this review and the clear reporting in the tables/figures.
Minor edits:
Ln 162-3 – Smart et al needs to be referenced appropriately

Ln 164 – “Studies were not excluded based on score for methodological quality.” One sentence to elaborate on why this was the case could help further clarify for the reader. Was this to increase study inclusion to be able to perform a MA?

Ln 196-7 – I2 needs to be superscripted x 2

Ln 216 – extra space between “reported” and “on”

Ln 238 – What does (%) mean? Is there a number that should precede the %?

Ln 269 – extra % in (46%%) needs removing

Ln 357 – Figure 3 appears to be referenced incorrectly and should be Figure 2.

Ln 372 – Figure 2 appears to be referenced incorrectly and should be Figure 3

Ln 387 – “meta regression” should be “meta-regression” or some consistent term

Ln 427-429- “A single study reported an initial sensation of leg numbness when one participant commenced BFRE, which did not persist beyond the first minute of exercise [39] (2 out of 16 hypertensive older women).” – would suggest changing wording to two participants since the study reported two adverse events in 16 participants, not just one participant.

Ln 452 – extra space present between “favored” and “dynamic”

Ln 466 – remove “the” before both

Ln 485 – needs period at the end

Ln 504 – need space between “40] and”

Ln 506 – Wilvebrand spelled wrong

Ln 514-524 – I appreciated this section in addressing the significant limitations in the literature and how that reflected in your analysis.

Ln 534 – Principals -> principles

Ln 545 – The article (Patterson, 2019) is now not referred to as a position stand, so this should be reflected in the manuscript

Ln 546 – space needs removing between recommending and BFR

Ln 555 – The article (Patterson, 2019) is now not referred to as a position stand, so this should be reflected in the manuscript

Ln 557 – needs a number before the percentage to inform reader of the exact % not using LOP as a standardization approach (also – you have used AOP prior – thoughts on sticking to one term throughout the manuscript?)

Ln 575-76 – additional spacing needs to be deleted, along with the comma between review and excluded

Ln 578 – stick with one variation of the word – comorbidities or co-morbidities and keep consistent. This has comorbidities but on 575 its co-morbidities.

May be good to mention https://www.ncbi.nlm.nih.gov/pmc/articles/PMC5374335/ even though there was no exercise control, it was RCO design and illustrates some of what you are talking about with respect to the at-risk populations (and was excluded from analysis based on your inclusion criteria) and shows some potentially negative changes in both the trained and untrained limb following 3 weeks of training in overweight individuals.

Ln 583 – “side” should have another word attached – ie “side events” if removing the preposition still doesn’t make sense.

Ln 610 – The article (Patterson, 2019) is now not referred to as a position stand, so this should be reflected in the manuscript; along with spacing between “tolerability” and “and”

---

## Round 0.2 · accepted · Accept

· Academic Editor

Accept

Thank you for your attention to detail in regards to the reviewers' comments and congratulations. Scotty

·

Basic reporting

The authors made the recommended adjustments to the manuscript and is suitable for publication.

Experimental design

The authors made the recommended adjustments to the manuscript and is suitable for publication.

Validity of the findings

The authors made the recommended adjustments to the manuscript and is suitable for publication.

Additional comments

The authors made the recommended adjustments to the manuscript and is suitable for publication.